# Alterations in Histone Methylation States Increased Profusion of *Lethal(2)-Essential-for-Life-Like (l(2)elf)*, Trithorax and Polycomb Genes in *Apis mellifera* under Heat Stress

**DOI:** 10.3390/insects15010033

**Published:** 2024-01-05

**Authors:** Ahmad A. Alghamdi, Yehya Z. Alattal

**Affiliations:** Department of Plant Protection, Chair of Engineer Abdullah Ahmad Bagshan for Bee Research, College of Food and Agriculture Sciences, King Saud University, Riyadh 11451, Saudi Arabia; alkhazim@ksu.edu.sa

**Keywords:** *A. m. jemenitica*, *A. m. carnica*, ChIP, *l2efl*, histone methylation, epigenetic

## Abstract

**Simple Summary:**

The honeybee subspecies *Apis mellifera jemenitica* is highly adapted to extreme summer temperatures. This adaptation can be explained by many anatomical, behavioral and molecular components. Basically, many genes are upregulated/silenced in response to stressors. Histone post-translational modifications (PTMs) represent an important gene regulation mechanism in insects. Here, we investigated how the methylation/demethylation of histone (H3) lysine (K4 and K27) impacts the profusion of *lethal(2)-essential-for-life-like (l(2)efl)*, histone methyltransferases (HMTs) and Polycomb genes in *A. m. jemenitica* and *A. m. carnica* under thermal stress. The results revealed significant alterations in histone methylation states leading to a higher profusion of *l(2)efl* and harmonized changes in HMTs and Polycomb genes in response to heat stress with a substantial variation between both honeybee subspecies. Apparently, histone PTMs can diminish the impact of thermal stress and increase the fitness of both honeybee subspecies. In conclusion, the methylation/demethylation of H3K4 and H3K27 is a key epigenetic mechanism in regulating *l(2)efl*, histone methyltransferases (HMTs) (*trx*) and Polycomb genes in *A. mellifera* exposed to heat stress.

**Abstract:**

Histone post-translational modifications (PTMs) represent a key mechanism in the thermal adaptation of the honeybee *Apis mellifera*. In this study, a chromatin immunoprecipitation assay and qPCR were employed to explore the changes in the methylation states of H3K4m2, H3K4m3, H3K27m2 and H3K27m3 associated with *l2efl* (ID: 72474, 724405, 724488), histone methyltransferases (HMTs) ((*trx*) and *PR-set7*) and Polycomb (*Pc*) and (*Su*(*z*)*12*) genes in *A. m. jemenitica* (tolerant subspecies) and *A. m. carnica* (susceptible subspecies) in response to heat treatment (42 °C for 1 h). The results revealed significant enrichment fold changes in the methylation/demethylation of most H3K4 and H3K27 marks at all targeted genes. These changes increased the profusion of *l2efl* (ID: 72474, 724405, 724488), histone methyltransferases (HMTs) (*trx)* and Polycomb (*Pc*) and *Su(z)12* and decreased the profusion of HMT (*PR-set7*) in both honeybee subspecies. The changes in the methylation enrichment folds of histone methyltransferases (HMTs) ((*trx), PR-set)* and Polycomb (*Pc*), *Su(z)12* genes demonstrate the well-harmonized coordination of epigenetic gene regulation in response to heat treatment. Compared to the control, the changes in the methylation enrichment folds of H3K4m3 at Polycomb *Su(z)12* were about 30× and 100× higher in treated *A. m. jemenitica* and *A.m. carnica,* respectively. Similarly, changes in the methylation/demethylation enrichment folds of HMT (*trx)* and Polycomb (*Pc*) and *Su(z)12* were 2–3× higher in *A. m. carnica* than in *A. m. jemenitica* after treatment (42 °C). It is evident that post-translational chromatin modification in both honeybee subspecies can diminish heat stress impact by (I) increasing the transcriptional provision of *l2efl* associated with survival and (II) increasing the silencing of genes associated with general cellular activities.

## 1. Introduction

The survival of honeybee colonies *Apis mellifera* L. is increasingly impacted by several ecological stressors associated with climate change [1,2,3,4]. For instance, heat shock under extreme summer temperatures is one of the key mortality factors of *A. mellifera* colonies [1,3,4,5]. With its eusocial structure, *A. mellifera* exhibit cooperative behaviors to reduce the impact of thermal stress, such as fanning, clustering and swarming [5,6,7,8,9]. In most cases, these common colony-based cooperative behaviors can only diminish the impact of short-term increases in daily or seasonal temperatures. Nevertheless, thermal adaptation varies significantly among different *A. mellifera* subspecies [4,10,11]. Subspecies from temperate zones such as *A. m. ligustica, A. m. mellifera* and *A. m. carnica* are very well adapted to temperate climates, yet these subspecies cannot withstand extreme summer temperatures when utilized in regions with extremely high temperatures [4,10,12]. In Europe, the dark European honeybee subspecies *A. m. mellifera* is the only subspecies that is highly adapted to a long flightless overwintering period under extremely low temperatures for several months (about −30 °C for about 6 months) [12]. 

However, the honeybee subspecies *A. m. jemenitica* belongs to the African honeybee lineage that occurs naturally in the subtropical areas of Africa and some countries of West Asia [10,13]. It demonstrates high thermal adaptation and reduced colony losses under extreme summer temperatures among all honeybee subspecies [4,11]. *A. m. jemenitica* is the endogenous honeybee of Saudi Arabia, which is a region with dry and long-term extreme temperatures exceeding 40 °C during the summer months [10,14,15]. Thermal adaptation in *A. m. jemenitica* may be related not only to cooperative colony-based behavior but also to intrinsic adaptive anatomical, behavioral and physiological traits in individual bees. Basically, the smaller body size and significant expression of heat shock proteins (*hsps*) are key aspects that explain the ability of *A. m. jemenitica* to survive under extreme temperatures [4,6,7,8,9,10,11,16,17,18,19,20,21]. At the molecular level, the expression of heat shock proteins (*hsps*) maintains the protein integrity of individual bees, which is a molecular mechanism that is well documented in *A. mellifera* [8,19,22,23,24]. Researchers reported a 150× higher day-long expression of HSP70 mRNA in *A. m. jemenitica* than in *A. m. carnica* under the desert conditions of Saudi Arabia [19]. 

Yet, *hsp* transcription might be associated with epigenetic inheritance [25,26,27]. Histone post-translational modifications (PTMs) represent the main epigenetic mechanism of thermal adaptation in *A. mellifera* [27,28,29]. These modifications include the methylation, acetylation, ubiquitination and phosphorylation of histone proteins leading to the upregulation/silencing of targeted genes [27,29,30]. Histone methylation is the addition of one/two/three methyl groups at the lysine (K) terminals of histones 3 (H3) and 4 (H4) [25,30]. This modification alters the methylation variants of histone states (H3K4, H3K9, H3K27, H3K36, H3K79 and H4K20) [25,30]. Yet gene activation or silencing is a cascade of steps that requires catalyzing enzymes [27,29,30]. Histone methyltransferases (HMTs) and histone demethylases (HDs) are two antagonistic enzymes that catalyze histone methylation (H3 and H4), and their abundance can be associated with histone modification [25,26,30]. The position and degree of histone methylation lead to the activation or inactivation of targeted genes. The histone marks H3K4me2 and H3K4me3 are transcriptional active marks, while H3K9me3, H3K27me3 and H4K20m3 represent suppressive chromatin marks [25]. However, Polycomb group proteins (PcGs) represent another epigenetic mechanism that maintains genes silencing [31,32,33]. These proteins usually act on histone and chromatin modifications, which impact the transcriptional regulation of target genes by creating repressive complexes [34]. For example, the trimethylation of histone H3 on lysine K27 is mostly associated with Polycomb protein complexes [35]. The expression levels of PcG mRNAs can be associated with changes in the transcriptional states of genes associated with thermal stress [34,35].

*Lethal(2)-essential-for-life-like (l(2)efl)* genes are epigenetically modulated in insects under biotic and abiotic stressors [36,37,38]. *l(2)efl* was first reported in *Drosophila melanogaster* with a high homology with heat shock protein 20 (HSP20). It is a constitutively expressed gene under normal conditions [39], and it shows differential expression under thermal, hunger and many other biotic and abiotic stressors in many insects [39,40,41,42,43,44]. 

In heat-stressed *Trichogramma chilonis,* the endoplasmic reticulum protein synthesis pathway was found to be active, involving the high upregulation of many genes mostly annotated as *lethal(2)-essential-for-life-like (l(2)efl)* and heat shock protein genes (hsps) [45]. The *lethal(2)-essential-for-life-like (l(2)efl)* genes stabilize protein intermediates by avoiding protein unfolding and the occurrence of insoluble aggregates; consequently, they can be suggested as ideal early signals for proteotoxic stressors [46,47,48]. In *A. mellifera,* the upregulation of the *lethal(2)-essential-for-life-like (l(2)efl)* genes plays an important role in ovary activation and cast differentiation [49,50]. In China*, A. m. ligustica* exhibited the upregulation of the *lethal(2)-essential-for-life-like (l(2)efl)* genes under mild winters [43]. The upregulation of the *lethal(2)-essential-for-life-like (l(2)efl*) genes (ID:724405; 724274; 724488) was also reported from within the mid-guts of heat-shocked bees at 45 °C [51]. In a recent study, changes in the expressions levels of *lethal(2)-essential-for-life-like* genes in *A. mellifera* were reported in response to different kinds of stressors, suggesting that these genes can be used as general evidence of cellular stress in honeybees [51]. 

The climate of Saudi Arabia with hot summers and dry winters presents an ideal place to study the impact of heat stress on different *A. mellifera* subspecies compared to *A. m. jemenitica*, which is the native honeybee subspecies. In this study, changes in histone methylation states (H3K4me2, H3K4me3, H3K27me2 and H3K27me3) were linked with the relative profusion of ChIPed-DNA (*lethal(2)-essential-for-life-like(l(2)efl)* genes, histone-lysine N-methyltransferase trithorax HMT (*trx*), histone-lysine N-methyltransferase PR-set7 (*PR-set7*), Polycomb (*Pc*) and Polycomb protein *Su(z)12* in *A. m. jemenitica* (thermally adapted subspecies) and *A. m. carnica* (thermally susceptible subspecies) after exposure to heat stress.

## 2. Materials and Methods

### 2.1. Honeybee Sample Preparation

The honeybee colonies were constructed as follows: four native honeybee colonies, *A. m. jemenitica*, were obtained from a certified source (Bee Research Unit, King Saud University, Riyadh, Saudi Arabia) representing the native honeybee subspecies (tolerant *A. mellifera* subspecies). Four other honeybee colonies, *A. m. carnica,* were assigned at the apiary of Penn State University (State College, PA, USA) representing the susceptible honeybee subspecies. Verification of subspecies affiliation was performed using morphometric methods (body size and pigmentation) according to standard procedures [10]. All colonies were standardized to form one brood box of 10 frames covered with adult bees including five brood frames. The colonies were treated alike in both localities (APIMONDIA guidelines of performance) [52]. The constructed honeybee colonies of both subspecies (4: *A. m. jemenitica* and 4: *A. m. carnica*) were used for sample collection. Adult nurse-age bees were samples from the constructed colonies. In total, 16 samples were taken: eight samples were taken from *A. m. jemenitica* (4 samples for treatment and 4 other samples for control) and another eight samples were taken from *A. m. carnica* colonies (4 samples for treatment and 4 other samples for control). Each sample had 10 bees (in total 80 bees from each honeybee subspecies were samples). Four samples (40 bees) from each subspecies were assigned for control and four samples (40 bees) were assigned for treatment. The samples assigned for the treatment were incubated at 42 °C for one hour (Treatment); the rest of the samples were incubated for 1 h at 34 °C (Control). After incubation, bee individuals of each sample were immediately frozen using liquid nitrogen. The bees’ thoraces of each sample were cut on a 100 mm Petri dish; then, necrotic materials and unwanted parts were removed. The tissues were then weighed and macerated.

### 2.2. Cross-Linking, Quenching and Chromatin Isolation

The prepared tissue of each sample was soaked in 1 mL of cross-linking solution. The cross-linking solution was prepared using formaldehyde and DMEM culture media (270 µL formaldehyde (37%) and 10 mL of DMEM: the final concentration of formaldehyde is 1% per every 40 mg of the thoraces tissues). After cross-linking, the tissues were incubated for 20 min at room temperature with moderate agitation. After cross-linking, the tissues were incubated for 15–20 min on a rocking platform at room temperature. The cross-linking step was quenched by the addition of 1 mL of 1.25 M glycine solution per every 9 mL of cross-linking solution. The tissues were then mixed and precipitated by centrifugation at 800 rpm for 5 minutes; then, the upper layer (supernatant) was removed and the tissues were rinsed three times with 10 mL of ice-cold phosphate buffer solution (PBS), once with centrifugation at 800 rpm for 5 min. The supernatant was then removed. At this step, eight cross-linked *A. m. carnica* samples: (4 for treatment (CT) and 4 for control (CC)) and eight cross-linked *A. m. jemenitica* samples (4 for treatment (YT) and 4 for control (YC)) were obtained. The samples representing the same subspecies and the same treatment were then pooled. At the end, four pooled samples were prepared (CT, CC, YT, YC), and each of them was homogenized. Chromatin isolation from each pooled sample was performed using a ChromaFlash^TM^ Chromatin Extraction Kit (P-2001, EpigenTek, Farmingdale, NY, USA). Chromatin isolation was performed following the kit instructions with a volume of 200 µL chromatin solution/sample. The isolated chromatin was sheared by the Episonic-2000 Sonication System, and then the concentration of the sheared chromatin was measured using a Qubit 4 Fluorometer^®^ (Thermo Fisher, Waltham, MA, USA). At the end of this step, we had four chromatin isolations: two from *A.m. jemenetica* (one for treatment and one for control) and two for *A. m. carnica* (one for treatment and one for control).

### 2.3. Chromatin Immunoprecipitation (ChIP)

For chromatin immunoprecipitation (ChIP) of the pooled samples (two samples for *A. m. jemenitica* (YC and YT) and two samples for *A. m. carnica* (CT and CC), K4me2, K4me3, K27me2 or K27me3 antibodies (6 µg) ((#A-4032; #A-4033; #A-4038; and #A-4039, respectively: EpigenTek, Farmingdale, NY, USA) were combined (in a 1.5 mL tube) with 12 µL ChIP assay beads, 3 µg chromatin solution and 500 µL ChIP assay buffer, respectively [53]. Non-immune IgG (6 µg) was used as the negative control. Then, the samples’ solutions were kept at room temperature for three hours with continuous agitation. After that, the beads were rinsed and the ChIPed-DNA was purified. The ChIPed-DNA was eluted using 60 µL of water. Quantities of ChIPed-DNA were determined using a Qubit 4 Fluorometer^®^ (Thermo Fisher, Waltham, MA, USA); for this purpose, 1 µL of purified DNA was used. DNA purification was performed using 10 µL of the sheared chromatin in the case of *A. m. carnica* and 20 µL in the case of *A. m. jemenitica*.

### 2.4. Primer Design

For primer design, sequences of the *A. mellifera lethal(2)-essential-for-life-like* genes (ID:724405; 724274; 724488); *A. mellifera* histone-lysine N-methyltransferase (HMT) (*trx*) (ID: 408716); *Apis mellifera* (HMT)-PR-set7 (*PR-set7*) (ID: 412027); *A. mellifera* Polycomb protein *Su(z)12*, mRNA (ID: 409170); *Apis mellifera* Polycomb (*Pc*), and mRNA (ID:725474*)* genes were downloaded from NCBI (Honeybee Genome Consortium). The sequences were then downloaded into Geneious^®^ Prime software version 2021.1.1 (https://www.geneious.com, accessed on 2 February 2023) for analysis and primer design (Table 1). 

### 2.5. Quantitative Polymerase Chain Reaction (qPCR)

The purified ChIPed-DNA of the pooled samples (CC, CT, YC and YT) was then subjected to a quantitative polymerase chain reaction (qPCR). The qPCR reactions were run using 1 µL of the ChIPed-DNA and specific primers for the relevant target genes (see below, Table 1). The Applied Biosystems 7500 Real-Time PCR (Applied Biosystems: ABI, Waltham, MA, USA) was used for qPCRs. Non-immune IgG DNA was used to calculate the enrichment fold. To determine enrichment efficiency (Input%), 5% and 10% of the Un-ChIPed DNA were used for *A. m. carnica* samples and *A. m. jemenitica* samples, respectively. The qPCR data (Ct) were then used to calculate enrichment folds (EF) and input% (I%) for each gene at different histone marks. EF = 2^ct(IgG-AB)^, I% = 2^ct (Input-AB)-(dilution factor)^ × 100%). Dilution factors of 20-fold (5%) for CC and CT and 10-fold (10%) for YC and YT were used to normalize input DNA to 100%. 

### 2.6. Statistical Analysis

The enrichment fold at the *(l(2)efl)* genes (ID:724405; 724274; 724488), the HMT (*trx*) (ID: 408716); the HMT-PR-set7 (*PR-set7*) (ID: 412027)*;* the *Polycomb Su(z)12* mRNA (ID: 409170); and the Polycomb (*Pc*) mRNA (ID:725474) were presented in means ± SD using GraphPad Prism 8.0.1. (www.graphpad.com, accessed on 10 September 2023). The mean comparisons of enrichment fold at targeted genes were performed using a pairwise *t*-test procedure (Benjamini, Krieger and Yekutieli (*p* < 0.05 and Q = 0.01)) of treated and control samples at each histone mark (H3K4m2, H3K4m3, H3K27m2, H3K27m3) [54]. Each methylation state was analyzed individually for each honeybee subspecies, and 4 *t*-tests were performed per gene/subspecies.

## 3. Results

### 3.1. Apis mellifera (l(2)efl) 

With regard to *A. mellifera lethal(2)-essential-for-life-like* genes (ID:724405; 724274; 724488), the results revealed significant enrichment fold changes in the methylation/demethylation states of all H3K4 and H3K27 marks at *l2efl* (ID: 72474, 724405, 724488) in both honeybee subspecies in response to treatment (Figure 1). The increased deposition of H3K4m2 and H3K4m3 was associated with a decreased deposition of H3K27m2 and H3K27m3 at *l2efl* (ID: 72474, 724405, 724488) (Figure 1), demonstrating an antagonistic response of H3k4 and H3K27 marks to heat treatment in both honeybee subspecies. Significant changes in the methylation states of gene repressing marks H3K27m2 and H3K27m3 were higher in *A. m. carnica* (representing susceptible honeybee subspecies) than in *A. me. jemenitica* (Tolerant honeybee subspecies), while changes in H3K4m2 and H3K4m3 were higher in *A. m. jemenitica* (Figure 1). All modification in the methylations states in both honeybee subspecies indicated a higher profusion of *l2efl* (ID: 72474, 724405, 724488) representing gene activation changes in response to heat treatment.

### 3.2. Apis mellifera histone-lysine N-methyltransferase (HMT) ((trx) and PR-set7)

The results revealed significant modification in the methylation/demethylation enrichment folds of H3K4m2, H3K4m3 and H3K27m2 at the histone-lysine N-methyltrans-ferase (HMT) (*trx*) gene (ID: 408716) in *A. m. carnica*. In *A. m. jemenitica*, changes in enrichments folds were significant for H3K4m3, H3K27m2 and H3K27m3 (Figure 2). The significance and dynamics of the methylation/demethylation changes at this gene (*trx*) mimics those of *l2efl* (ID: 72474, 724405, 724488) at most histone methylation marks (H3K4m2; H3K4m3; H3K27m2; H3K27m3) (Figure 2). Although variation in the methylation/demethylation states was significant in both honeybee subspecies, the variation was about 10× higher in *A. m. jemenitica* than in *A. m. carnica* (Figure 2). The highest changes occurred at gene inactivation mark H3K27 in both honeybee subspecies. The variations in the methylation/demethylation enrichment folds at PR-set7 (*Apis mellifera* HMT- PR-set7 (*PR-set7*) ID: 412027) were antagonistic to the changes that occurred at the histone-lysine N-methyltransferase trithorax (*trx*) (ID: 408716). The changes were significant for H3K4m2 and H3K27m2 at *PR-set7* in *A. m. carnica* in response to treatment (Figure 2). In *A. m. jemenitica*, changes in the methylation/demethylation enrichment folds at *PR-set7* were insignificant except for H3K4m2 only (Figure 2). The main changes occurred at H3K27m2 for the HMT gene.

### 3.3. Apis mellifera Polycomb (Pc) and (Su(z)12)

The results showed that the changes in the methylation/demethylation enrichment folds of H3K4 and H3K27 at Polycomb genes (*Pc*) and *PR-set7*) were much higher compared to *Apis mellifera* histone-lysine N-methyltransferase (HMT) ((*trx*) and *PR-set7*) genes in response to heat treatment in both *A. mellifera* subspecies (Figure 2).

At Polycomb (*Pc*) gene (ID: 725474), the variation in the methylation/demethylation enrichment folds was highly significant in *A. m. carnica* for H3K4m3, H3K27m2 and H3K27m3 in response to treatment (Figure 2). In *A. m. carnica*, the variation in the methylation/demethylation enrichment folds of H3K4 at Polycomb (*Pc*) was 20× more in the treated samples than in the control samples (Figure 2). In *A. m. jemenitica*, the methylation enrichments folds varied significantly at H3K4m2, H3K27m2 and H3K27m3 (Figure 2). Generally, the changes in the methylation/demethylation states were higher in *A. m. carnica* than in *A. m. jemenitica.*

The changes in the methylation/demethylation states at the Polycomb *Su(z)12* (ID: 409170) were the highest among all other changes in both honeybee subspecies. At Polycomb *Su(z)12* (ID: 409170), the changes in the methylation/demethylation enrichment folds were significant for the states K4m2, K4m3 and K27m2 in *A. m. jemenitica* and for K4m3 and K27m2 in *A. m. carnica* in response to heat treatment (Figure 2). The variation in the methylation/demethylation enrichment folds of H3K4m3 was about 30× and 100× higher in *A. m. jemenitica* and *A.m. carnica*, respectively, in response to treatment. 

## 4. Discussion

The study demonstrated an increased profusion of *l2efl* (ID: 72474, 724405, 724488), HMT (*trx)*, (*Pc*) and *Su(z)12* associated with the methylation/demethylation states of histone H3 lysine K4 and K27 in both honeybee subspecies after exposure to heat treatment (42 °C for 1 H) and a decreased profusion of *PR-set7* with increased methylation of the gene silencing marks K27m2 in both honeybee subspecies. These changes in the methylation/demethylation enrichment folds of different histone H3 lysine K4 and k27 marks (H3K4m2, H3K4m3, H3K27m2, H3K27m3) suggested a key post-translational chromatin modification of the *l2efl* (ID: 72474, 724405, 724488) in response to heat stress, demonstrating a layer of epigenetic regulation of *l2efl* (ID: 72474, 724405, 724488) transcription in *A. mellifera*. The changes in the methylation/demethylation enrichment folds were generally significant in *A. m. jemenitica* (honeybee subspecies representing thermal adaptation) and *A. m. carnica* (a subspecies of temperate origin). Still, the enrichment fold values at all histone marks (especially at gene silencing mark H3K27) might demonstrate potentially higher transcriptional levels *of l2efl* (ID: 72474, 724405, 724488) in *A. m. jemenitica* compared to *A. m. carnica* even under control conditions and higher response (sensitivity) in *A. m. carnica* in response to heat stress. Modification of the chromatin methylation states at H3K4 and H3K27 was previously reported to increase the profusion of HSP70 in both honeybee subspecies under desert conditions, demonstrating a higher response variation in *A. m. carnica* due to treatment (42 °C) [28]. 

The histone methyltransferase HMT (*trx)* catalyzes histone lysine methylation at H3K4 and leads to gene activation. The higher methylation enrichment folds of H3K4m2 and H3K4m3 and demethylation of H3K27m2 and H3K27m3 at HMT (*trx*) indicate a significant post-translational modification and increased profusion of HMT (*trx*) in response to heat stress in both honeybee subspecies. These variations in HMT *(trx)* methylation/demethylation enrichment folds at H3K4, H3K4, H3K27 and H3K23 match the higher profusion of *l2efl* (ID: 72474, 724405, 724488). On the other hand, the histone methyltransferase HMT (*PR-set7)* catalyzes the methylation of H4K20 leading to gene suppressing, demonstrating antagonistic activity to HMT (*trx*). The decreased profusion of *PR-set7* in response to treatment, mainly in *A.m. carnica,* indicates higher methylation activities at H3. This remolding of H3K4 at HMT *(trx)***,** HMT *(PR-set7)* and *l2efl* (as well as *HSP70* [28]) demonstrated a high harmonized coordination of gene activation in response to heat treatment. The relatively higher profusion of *l2efl* (ID: 72474, 724405, 724488) and HMT (*trx*) in *A. m. carnica* after treatment is expected. *A. m. carnica* is a susceptible honeybee subspecies with very low survival rates under hot and dry conditions; consequently, its heat-stress threshold temperature can be much lower compared to *A. m. jemenitica*. On the other hand, *A. m. jemenitica* is one of the honeybee subspecies most adapted to extreme temperatures with many behavioral, anatomical and molecular components of adaptation [10]. This may lead to slower or delayed heat survival response under 42 °C or to an earlier/continuous accumulation of target gene products (*l2efl* (ID: 72474, 724405, 724488)) even under control conditions in *A. m. jemenitica*. In a previous long-term study comparing the survival rates of honeybee colonies of *A. m. jemenitica* and *A. m. carnica* under desert conditions of Saudi Arabia, it was reported that most *A. m. carnica* colonies (92%) die within the first and second seasons due to extreme summer temperatures compared to only 46% in *A. m. jemenitica* [4]. Still, the changes in histone marks in response to heat shock were much higher in *D. melanogaster* (isothermic insect) than in *A. mellifera* (heterothermic insect) [55]. 

The Polycomb genes (*Pc*) and *Su(z)12*) are examples of chromatin-based regulatory genes that silence the transcription of target genes by the trimethylation of H3K23 and H3K9, respectively [56,57]. In this study, heat treatment increased the profusion of Polycomb gene (*Pc*) ID: 725474) by an increased methylation of K4m3 and demethylation of K27m2 and K27m3 in *A. m. carnica*. This increased profusion of Polycomb gene (*Pc*) ID: 725474) interacts with H3K27 acetylation genome-wide, causing gene silencing at specific loci [58]. Consequently, heat stress can be associated with a lower transcription in *A. m. carnica.* In *A. m. jemenitica*, changes in the methylation enrichment folds of K4m2, K27m3 and K27m3 were not similar, indicating a milder impact on Polycomb gene (*Pc*) profusion, which might be associated with the higher fitness and survival rates of *A. m. jemenitica* under extreme temperature. The changes in the methylation/demethylation states of K4m2, K4m3 and K27m2 at Polycomb *Su(z)12* were the highest in both honey bee subspecies, indicating a higher response of *Su(z)12* leading to suppression. Generally, this gene suppression impairs many physiological activities in insects [59]. It is likely that post-translational chromatin modification in both honeybee subspecies aimed at diminishing heat stress by (I) increasing the transcriptional provision of genes associated with survival such as *l2efl* and (II) increasing gene silencing at other targeted genes associated with general cellular activity. It might be recommended to explore colony-wise variations associated with HPTM at these specific genes under heat stress. 

In conclusion, this study presents evidence on epigenetic changes at targeted genes (*l2efl*, HMT (*trx*), (*Pc*) and *Su(z)12*) associated with heat treatment in tolerant and susceptible *Apis mellifera* subspecies. Furthermore, it can be concluded that epigenetic changes in response to heat treatment are also evident in *A. m. jemenetica*, which is a highly adapted honeybee subspecies to heat stress.

## Figures and Tables

**Figure 1 insects-15-00033-f001:**
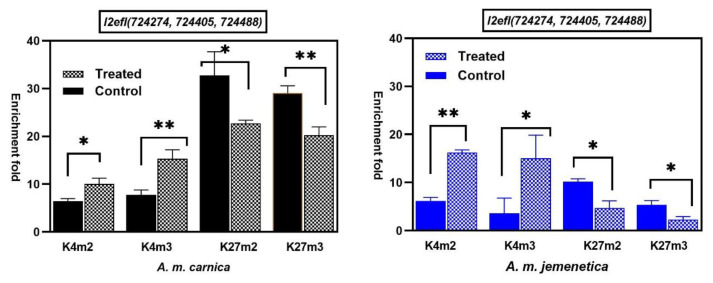
Enrichment-folds’ comparison (means ± SD) of ChIP-qPCR for different histone marks (K4m2, K4m3, K27m2, K27m3) at: *lethal(2)-essential-for-life-like (l(2)efl)* (*A. mellifera*) (genes IDs: 724274; 724405; 724488) in the native tolerant honeybee subspecies of the Arabian Peninsula *A. m. jemenitica* and the susceptible *A. m. carnica* after exposure to 42 °C for one hour. Antibodies of histone marks associated with gene activation (H3K4me2 and H3K4me3) and histone marks associated with gene repression (H3K27me2 and H3K27me3) were used to precipitate sonicated chromatin of each sample. Enrichment fold (EF) was calculated using DNA from non-immune IgG as negative control. EF = 2^ct(IgG-AB)^. Pairwise comparison of means was performed using GraphPad Prism 8.0.1. (www.graphpad.com, accessed on 10 September 2023). (*: 0.01 ˂ *p* ≤ 0.05, **: 0.001 ˂ *p* ≤ 0.01).

**Figure 2 insects-15-00033-f002:**
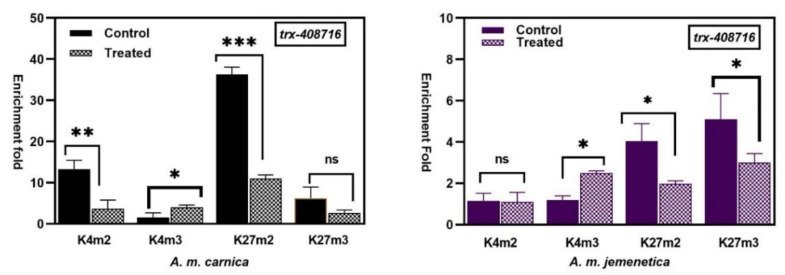
Enrichment folds’ comparison of ChIP-qPCR (means ± SD) of histone marks (K4m2, K4m3, K27m2, K27m3) at *A. mellifera* histone-lysine N-methyltransferase (*trx*) gene ID: 408716; *Apis mellifera* histone-lysine N-methyltransferase pr-set7 (*PR-set7*), mRNA, gene ID: 412027; *A. mellifera* Polycomb protein *Su(z)12*, mRNA gene ID: 409170; *Apis mellifera* Polycomb (*Pc*), mRNA, gene ID:725474 in the native honeybee subspecies of the Arabian Peninsula *A. m. jemenitica* and *A. m. carnica* after exposure to 42 °C for one hour. Activation histone marks antibodies (H3K4me2 and H3K4me3) and silencing marks (H3K27me2 and H3K27me3) were used to precipitate sonicated chromatin. Enrichment fold (EF) was calculated using DNA from non-immune IgG as negative control. EF = 2^ct(IgG-AB)^. Pairwise comparison of means was performed using GraphPad Prism 8.0.1 (www.graphpad.com, accessed on 10 September 2023), (ns: non-significant; *: 0.01 ˂ *p* ≤ 0.05, **: 0.001 ˂ *p* ≤ 0.01, ***: *p* ≤ 0.001).

**Table 1 insects-15-00033-t001:** Gene names (gene ID), gene subcellular location/length and designed primer sequence for *l(2)efl*, histone methyltranferases ((*trx*), (*PR-set7*)) and Polycomb genes (*Pc*), (*Su(z)12*) used in the present study.

Gene ID/Gene Name	Location (LG2)/Length	Primers
*(l(2)efl)* (*A. mellifera*) ID:724405	(LG2):4831304..4832577 /1274nt	F-TGCGACATCGATCAAGCGTCC R-TTGCGCATCGCACGGTTTCC
*(l(2)efl)* (*A. mellifera)* ID:724488	(LG2):4837466..4838343 /878nt	F-ACCTTGGGGTGAACTTCTGCG R-TCCCCTCGACGACAACACAC
*(l(2)efl)* (*A. mellifera*) ID:724274	(LG2):4823146..4824181 /1063nt	F-TCACCGAGCCGATTGGAGTTATGT R-AACTGCCTCTGTCACCACGAAAC
*A. mellifera* HLMT (*trx*) ID: 408716	(LG2):4633633..4650053, complement	F-TGCAGCTAGATTCATTAATCATTCATGR-CATGGAATCTTGATATCCTCGAAAG
*A. mellifera* Polycomb protein *Su(z)12*, ID: 409170	(LG10)11912175..11918682	F-ATGCTCTGCCCAAGCAACTATTACG R-CGGAACCTCCATCTTGTTACATAAA
*Apis mellifera* HLMT- PR-set7 (*PR-set7*), ID: 412027	(LG8) (1813753..1816546)	F-TGGTAAAGGTCGTGGAATAGTAACA R-AGTTTCTGCTGTTGCATCAACGC
*Apis mellifera* Polycomb (*Pc*), ID:725474.	(LG3) (8393126..8397763, complement)	F-ACAACGTCAAAGCAATGACAAATTA R-AGCTGCTCCAAAATATATGTTCACCG

## Data Availability

Data are available in tables and figures.

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
