# Peer review of "Alterations in Histone Methylation States Increased Profusion of Lethal(2)-Essential-for-Life-Like (l(2)elf), Trithorax and Polycomb Genes in Apis mellifera under Heat Stress"

_insects, 2024, doi:10.3390/insects15010033_

Round 1

Reviewer 1 Report

Comments and Suggestions for Authors

Review of Insects manuscript:  Alterations in histone methylation states increased profusion of lethal(2)-essential-for-life-like (l(2)elf), trithorax and Polycom genes in Apis mellifera under heat stress.

The authors use molecular techniques to study epigenetic changes in honeybees exposed to heat treatments.  They specifically investigate histone posttranslational modifications associated with certain key genes using ChIP-qPCR.  They find evidence that there are changes to HPTMs in the investigated genes.  However, there are not strong differences in the HPTMs associated with these genes in different bee subspecies. 

Overall, I found this to be a reasonable study.  The study is highly technical in presentation and a bit dense to read.  It is also rather narrowly focused to just a few genes, as opposed to a full genome level study (e.g., ChIP-Seq) that might have provided considerably more information.  So the study is relatively small in scale.  But it does provide new information on epigenetic changes associated with environmental variation in social insects.  So I only have minor comments at the level presented. 

Minor Comments

The authors can’t show that the HPTMs actually cause changes in expression of the genes (as suggested in the title).  They can only show that these HPTMs associated with the focal genes do change under experimental conditions. 

The entire Introduction seemed to be presented as only one paragraph.  This should be broken up into multiple different paragraphs to make it more readable. 

The authors could do a better job explaining why they chose to study the specific genes that they investigated.  They make mention of this at the end of the Introduction, but it should be explained in more detail why these genes are particularly interesting targets for this particular analysis. 

The research team does a good job in describing their sampling scheme.  But I was still not completely sure of how many samples were used for the molecular analyses.  Was ChIP done on all 80 samples?  And QPCR also conducted on all 80 samples? 

The authors say that they collected bees from 4 distinct colonies per species.  It would be interesting to test if individuals from different colonies showed different patterns of ChIP-qPCR.  This is somewhat important because combining data across colonies can be problematic if colonies themselves show differences. 

Comments on the Quality of English Language

Some English editing would be helpful, but the quality is reasonable overall.  

Reviewer 2 Report

Comments and Suggestions for Authors

insects-2765366 Reviewer comments

Manuscript insects-2765366: Alterations in histone methylation states increased profusion of lethal(2)-essential-for-life-like (l(2)elf), trithorax and Polycom genes in Apis mellifera under heat stress.

The manuscript is very interesting. The authors investigated impact of methylation/demethylation of histone (H3) lysine (K4 and K27) to the profusion of lethal(2)-essential-for-life-like (l(2)efl), histone methyl transferases (HMT) and Polycom genes in A. m. jemenitica and A. m. carnica under thermal stress. Results revealed significant alteration in histone methylation states leading to higher profusion of l(2)efl and a harmonized changes in HMT and Polycom genes in response to heat stress with substantial variation between both honeybee subspecies. Apparently, histone of post-translational modifications (PTM) can diminish the impact of thermal stress and increase survivability in both honeybee subspecies. Methylation/demethylation of H3K4 and H3K27 is a key epigenetic mechanism in regulating l(2)efl, histone methyl transferases HMT (trx) and Polycom genes in A. mellifera exposed to heat stress.

The uniqueness of the text is 90% by antiplagiarism.net

The experimental and statistical methods are correct.

The English is almost good but need some correction by native speaker..

There are some mistakes and comments:

The manuscript do not have line numbers, it makes difficulties in localization of misspelling positions.

1) Throughout the text -  jemenetica - should be - jemenitica.

2) In the abstract - subspeceis - should be - subspecies.

3) In the abstract - demostrated - should be - demonstrated.

4) In the Materials and Methods - Thermotolernt - should be - Thermo-tolerant.

5) In the Materials and Methods - Thermofisher and Epigentek  - should be - Thermo Fisher and EpigenTek.

6) In the Materials and Methods - there are two e - e. Primer design and e. Statistical Analysis. Please correct it.

7) In he Materials and Methods -  (4 for treatment (CT) and 4 for control (CT);) - should be - (4 for treatment (CT) and 4 for control (CC);)

8) In the Results - subspeceis - should be - subspecies.

9) In the Results - demethylatin - demethylation.

8) In the Results 3.3 - jemnetica - should be - jemenitica.

10) In the Results 3.3 - histone methyle tranferases - should be - histone methyltransferases.

11) In the Results 3.3 -  N-methyltrans-ferase - should be -  N-methyltransferase.

12) In Discussion - canica - should be - carnica.

13) In Discussion - thermosusceptible - should be - thermo-susceptible.

14) In Discussion - trimelylation - should be - trimethylation.

15)  In Discussion - Deputyship -  - Deputyship.

16) I do not understand what aim of your experiment and why you compared A.m.jemenitica with A.m.carnica. There are other tropical bees adapted to hot climate.

17) After the sentence - Subspecies from temperate zone such  as A. m. ligustica and A. m. carnica are very well adapted to temperate climates, yet these  subspecies cannot withstand extreme summer temperature when utilized in regions with  extreme high temperature [4,6,12] 

- add additional sentence - In Europe, the dark European honey bee subspecies A.m.mellifera is only subspecies highly adapted to extremely low, about minus 30 Celsius degree, temperature and long, about 6 months, flightless wintering period (Ilyasov et al., 2020).

Add to the References:

Ilyasov, R.A.; Lee, M.-L.; Yunusbaev, U.B.; Nikolenko, A.G.; Kwon, H.-W. Estimation of C-derived introgression into A. m. mellifera colonies in the Russian Urals using microsatellite genotyping. Genes and Genomics 2020, 42 (9), 987–996. doi: 10.1007/s13258-020-00966-0

Please improve the manuscript according to the above comments.

Comments are also highlighted in the text.

Comments on the Quality of English Language

Minor editing of English language required.
